# Multidimensional Spatial Match of Hierarchical Healthcare Facilities Considering Floating Population: A Case of Beijing, China

**Xingfei Cai** [1], **Hao Wang** [1,*], **Xiaogang Ning** [1], **Qiyong Du** [2] **and Peng Jia** [3,4,*]

1   Institute of Photogrammetry and Remote Sensing, Chinese Academy of Surveying & Mapping,
    Beijing 100036, China; 15071335295@163.com (X.C.); ningxg@casm.ac.cn (X.N.)
2   Guangzhou Urban Planning Survey and Design Institute, Guangzhou 510060, China; duqiyong@whu.edu.cn
3   School of Resource and Environmental Sciences, Wuhan University, Wuhan 430079, China
4   International Institute of Spatial Lifecourse Epidemiology (ISLE), Wuhan University, Wuhan 430079, China
*   Correspondence: wanghao@casm.ac.cn (H.W.); jiapengff@hotmail.com (P.J.)

**Abstract:** Good health and well-being are key to achieving the main goals of the UN Sustainable Development Goals (SDGs), especially after the outbreak of the COVID-19 epidemic. What is a concern for both government and society is how to understand the spatial match of hierarchical healthcare facilities and residential areas in terms of quantity and capacity, to meet the challenges of various diseases and build a healthy life. Using hierarchical healthcare data and cellphone signaling data in Beijing, China, we used the kernel density estimation, a bivariate spatial autocorrelation model, and a coupling index to explore the spatial relationships between hierarchical healthcare facilities and residential areas. We found large numbers of both healthcare facilities and residential areas in the urban center, and small numbers of both at the urban edge. The hospitals and designated retail pharmacies in the densely populated areas do not have enough capacity to meet the need of the population. In addition, the capacity of primary healthcare institutions can meet people's needs. Our findings would serve as a reference for urban planning, optimization of hierarchical healthcare facilities, and research on similar themes.

**Keywords:** hierarchical healthcare facility; spatial match; designated retail pharmacy; cellphone signaling data

## 1. Introduction

Ensuring healthy lives and promoting well-being at all ages are essential to the UN Sustainable Development Goals (SDGs) and targets. Target 3.3 aims to eliminate and fight epidemics. Currently, we are still faced with the problem of the uneven distribution of healthcare resources and defective healthcare security among different groups of people. The COVID-19 pandemic has exacerbated these problems. Since its outbreak, we have all been living in hard times. Never has it been so evident that life is a matter of survival, in addition to growth and flourishing. The UN Sustainable Development targets require new ways of conceptualizing and ensuring healthy lives, defining it not only as infrastructural construction but also as a rational distribution. Thus, understanding the spatial match of healthcare facilities and residential areas in terms of quantity and capacity is of great significance to responding to the ever-changing epidemic, improving the hierarchical healthcare system, and achieving the UN Sustainable Development Goals.

As an important part of urban public service facilities, the spatial match of healthcare facilities with residential areas is crucial to people's health. Many studies have explored from different sides how healthcare facilities are distributed and matched in space. As for research contents, geographers are concerned with healthcare facilities' spatial patterns [1], location allocation [2], and accessibility [3–5] to explore the geographical phenomena

and knowledge. Planning scholars have identified the current situation of healthcare facilities from the views of function evaluation [6] and service level [7] to cope with future planning. For example, the problems of space layout features and existence have been analyzed [8] and the spatial layout and functional differences of Hong Kong have been evaluated [9]. Regarding research themes, vulnerable groups [10], such as the elderly [11] and children [12], marginal areas such as poor countries [13] and villages [14–16], healthcare emergencies [17,18], and healthcare in response to disaster emergency services [19,20], as well as other special healthcare services in emergencies, are involved. For the above objects, some research has focuses on the diversified layout and capacity of healthcare facilities, and the opportunities for such groups and regions to obtain basic healthcare services under the condition of an unbalanced distribution of healthcare resources and multi-modal transport of healthcare facilities [21,22]. For instance, the evolution trend of spatial correlation between the elderly population number and the healthcare number has been investigated [23]. The transportation conditions and emergency responses accessibility of critical public service sectors under normal conditions and multiple flood scenarios have been evaluated [24]. In summary, both practical and theoretical investigations of spatial allocation within the previous research have revealed its crucial effects on healthy living, tremendously improving our understanding of the complicated spatial relationships between healthcare facilities and residential areas.

However, there are still two aspects to be improved. On the one hand, the spatial pattern of designated retail pharmacies, as an important part of the hierarchical healthcare system, should be concerned. Target 3.8 points out that universal health protection must be achieved, everyone must enjoy high-quality basic healthcare services, and everyone must have access to safe, effective, high-quality, and affordable basic medication. To achieve this goal, multi-level healthcare facilities, including hospitals, primary healthcare institutions, and designated retail pharmacies, should be considered to construct a system of hierarchical healthcare. However, there is little research on designated retail pharmacies, which makes it unclear how the spatial patterns of designated retail pharmacies impact the system, as designated retail pharmacies play an important role in protecting the physical and mental health of residents, especially in China's healthcare insurance system reform [25]. On the other hand, the floating population should be considered to more precisely depict the spatial match of hierarchical healthcare facilities with residential areas, for everyone has the right to health. In terms of capacity, the relationship between healthcare facilities and residential areas is more dependent on the resident population [26]. As a result, the spatial match of hierarchical healthcare facilities might blur the effects of the floating population in terms of capacity, leading to inflated estimations. What is more, as an important force in the rapid process of urbanization, the floating population is also a vulnerable group of basic public services. However, their health problems are often ignored by society [27].

Our study aims are twofold: (1) to portray the spatial patterns of hierarchical healthcare facilities, which include hospitals, primary healthcare institutions, and designated retail pharmacies; (2) to examine the spatial match between hierarchical healthcare facilities and residential areas in terms of quantity and capacity, taking floating population into account. We analyzed the spatial match of quantity by calculating the number of hierarchical healthcare facilities and residential areas. Then, we measured the spatial match of capacity using the total population and the number of patients receiving treatment. We also used the distribution data of hierarchical healthcare facilities and cellphone signaling data to explore the spatial distribution of hierarchical healthcare facilities and their spatial match with residential areas in terms of quantity and capacity. This study provides some suggestions for relevant government departments to identify areas lacking healthcare facilities and to improve spatial distributions of healthcare, thus helping to achieve the goal of equal access to healthcare.

## 2. Materials and Methods

### 2.1. Methodological Flow

The overall workflow of our study is presented in Figure 1. We used multiple types of data to study the spatial relationship between hierarchical healthcare facilities and residential areas in quantity and capacity. The steps are as follows: first, based on the value of kernel density estimation, the kernel density maps of hierarchical healthcare facilities are drawn, and the density images are divided into 5 categories according to the method of natural breaks. Secondly, we use bivariate spatial autocorrelation to analyze the spatial relationship in quantity and perform the significance test. Lastly, to further explore the spatial match in capacity, the coupling index is calculated from the street scale by using the geographical concentration and the coupling index. Taking into account the needs of urban management, we analyze the spatial match from the street level and multiply the annual average number of patients receiving treatment from different healthcare facilities by the number of healthcare facilities as the capacity of this street.

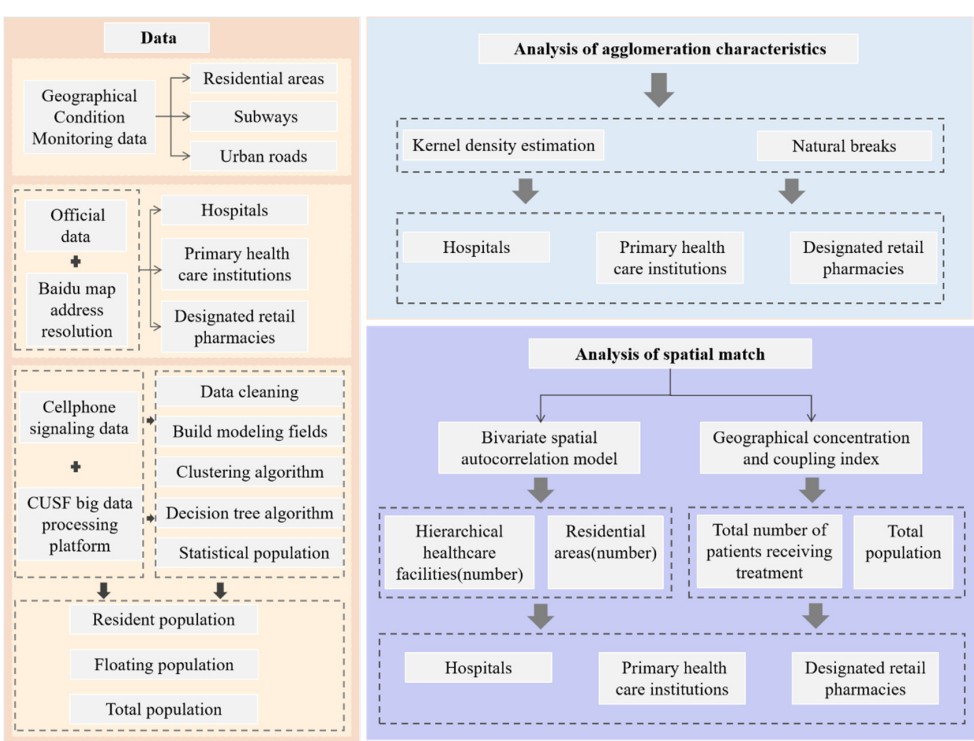

**Figure 1.** The flowchart of the research.

### 2.2. Study Area

The Healthy Beijing 2030 Planning Outline points out that the contradiction between the supply and demand of health services in Beijing is still contradictory, and problems such as the unbalanced distribution of health service resources and relatively weak primary service capacity have not been fundamentally resolved. It is urgent to promote the construction of health in Beijing at an all-around, multi-level, and high level. In addition, as one of the most concentrated cities of floating population in China, the uneven distribution of floating population is also a major concern in Beijing's social governance. Therefore, Beijing, with 16 districts, was selected as the study area, which is divided into five regions: core area (Dongcheng and Xicheng), sub-center (Tongzhou), central city (Chaoyang, Haidian, Fengtai, and Shijingshan), suburban new town (Mentougou, Fangshan, Shunyi, Changping, and Daxing), and outer suburban new town (Huairou, Pinggu, Miyun, and Yanqing) [28], as shown in the Figure 2. The core area, sub-center, and central city are defined from the perspective of urban function, while the suburban new town and outer suburban new town are defined from the perspective of spatial distance. The core area, which is called the

functional core area of the capital, is the core bearing area of the national political center, cultural center, and international communication center. The sub-center is proposed to adjust the spatial functional pattern of Beijing, control the diseases of big cities, and expand the new space for development. It is also necessary to promote the coordinated development of Beijing-Tianjin-Hebei and explore the optimal development mode of densely populated areas. The central city refers to the main areas that relieve noncapital functions. The suburban new town is defined as an area with a radius of 20–30 km from the core area. The outer suburban new town is defined as an area with a radius of 50–70 km from the core area [29].

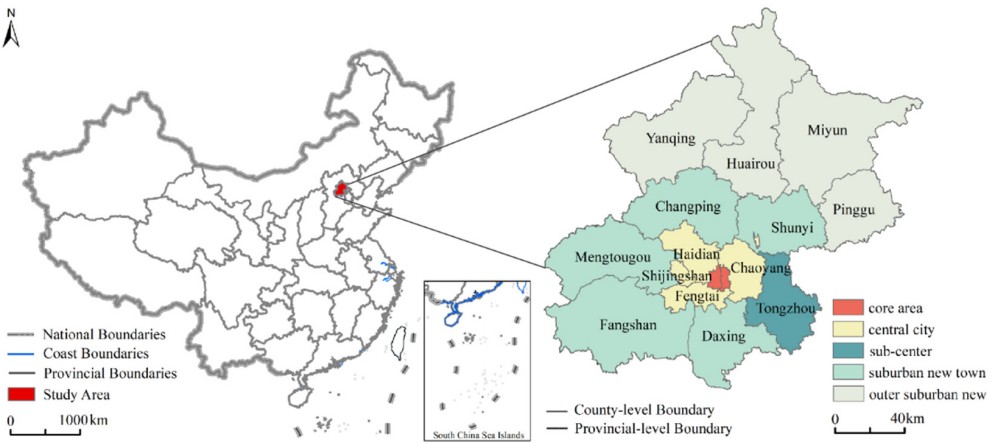

**Figure 2.** The location of the study area.

### 2.3. Data Sources

In this study, we used three types of data to analyze the spatial patterns and spatial match of hierarchical healthcare, including Geographical Condition Monitoring (GCM) data: residential areas, subways, and urban roads; and official data: hierarchical healthcare facilities including hospitals, primary healthcare institutions, and designated retail pharmacies. Additionally, cellphone signaling data were used for extracting resident population and floating population. All data were from 2018.

#### 2.3.1. Geographical Condition Monitoring Data

Geographical Conditions Monitoring (GCM) is a dynamic monitoring of the territory of the country from two sides by comprehensively utilizing modern surveying and mapping technologies such as the Global Navigation Satellite System (GNSS), Remote Sensing Technology (RS), and Geographic Information System Technology (GIS). The first is to identify the status and spatial distribution of natural and cultural geographical features, such as rivers and lakes, forests, grassland, road network, and urban layout. The second is to analyze their change in quantity and frequency, distribution characteristics, regional differences, and trends. GCM started in 2015 and has been updated annually since then. We mainly introduce how to obtain residential areas, subways, and urban roads. The data of residential area are provided by the city management department, and vectorized and verified using big data and field surveys. The data of subways and urban roads are provided by the Transportation Committee, and then vectorized and verified by using high-resolution remote sensing images and field investigations [30]. GCM is organized by the Ministry of Natural Resources for data production, quality inspection, and acceptance, with quality ensured by internal interpretation and field verification.

#### 2.3.2. Official Data

According to the National Health Service System Planning Outline (2015–2020) and considering the increase in the demand for the insured person to purchase healthcare insurance drugs nearby, the hierarchical healthcare facilities are reclassified into three categories,

including hospital, primary healthcare institution, and designated retail pharmacy. The hospital is the main body of China's healthcare service system, which undertakes the role of providing basic healthcare services, diagnosis, and treatment of critical diseases. It also provides medical research, medical teaching, and undertakes statutory and government-designated public health services, emergency medical rescue, foreign aid, among others. The primary healthcare institution provides prevention, healthcare, and health education. It also provides basic public health services, treatment services for common and frequently occurring diseases, and rehabilitation and nursing services for some diseases. The designated retail pharmacy is the product of the medical insurance system, which is approved by the labor security administrative department and determined by the medical insurance agency to provide prescription dispensing and nonprescription drug retail services for the insured [31]. The distribution data of hierarchical healthcare facilities were obtained from the Beijing government data resource network (https://data.beijing.gov.cn/ (accessed on 10 September 2021)). The data provide detailed actual information for any given point location, including its name, address, and facility category. We converted text information into latitude and longitude coordinate information through the Baidu map address resolution, and the Baidu coordinate system was converted into the China Geodetic Coordinate System 2000 (CGCS2000). Then, the data for each area were sampled and checked to ensure the accurate spatial position of facility point data. The obtained data were composed of 10,006 records. We used H1–H6 to indicate the category of hospitals, P1–P4 to indicate primary healthcare institutions, and D1 to indicate designated retail pharmacies. The classification system of the healthcare service facility is shown in Table 1.

**Table 1.** Categories of healthcare facilities.

| Data Category | Serial | Category Name | Amount |
|---|---|---|---|
| Hospitals | H1 | General hospitals | 724 |
| | H2 | TCM hospitals | |
| | H3 | Combined traditional and western medicine hospitals | |
| | H4 | National hospitals | |
| | H5 | Specialized hospitals | |
| | H6 | Nursing homes | |
| Primary healthcare institutions | P1 | Community health service centers | 8509 |
| | P2 | Community health service stations | |
| | P3 | Health centers | |
| | P4 | Outpatient departments and clinics | |
| Designated retail pharmacies | D1 | - | 773 |

### 2.3.3. Cellphone Signaling Data

The cellphone signaling data were obtained from China Unicom operators, which cover the whole Beijing area, including 24 h signal trajectory recording. Cellphone signaling data have the characteristics of wide coverage, large data samples, and strong timeliness [32]. Many research results have been achieved in the spatial and temporal distribution of population [33], which provides the possibility for the matching study of hierarchical healthcare facilities and residential areas in terms of capacity. We introduce the data from three aspects:

- Data preprocessing. Due to a large amount of data, ordinary software is not competent for processing. Therefore, the China Unicom Smart Footprint (CUSF) big data processing platform is used for saving, reading, and preprocessing the data in our study, which is an advanced big data processing framework based on the Spark cluster of the Hadoop architecture. The noise of the raw signaling data mainly includes the time errors and data that cannot effectively track the International Mobile Subscriber Identity (IMSI) number. The base station parameters are used to clean the noise in the signaling data to obtain the correct signaling generation location [34].

- Population type identification. To identify population types, four steps are set. First, we construct the stay and behavior characteristics in both time the dimension and spatial dimension. In the time dimension, the number of signaling items, stay time, stay days, and stay months of each user in the corresponding scene area is counted. In the spatial dimension, the location of the user is recorded. Secondly, we use two-step and K-means clustering algorithms to cluster the crowds. Thirdly, the decision tree algorithm is used to classify the crowds to the resident population and the floating population. Finally, the resident population and the floating population in the area are counted according to the population identification tags. The algorithm mainly determines where the user resides. There are two core judgment logics: one is that the user has at least two consecutive signaling events in the same base station and its vicinity; the other is the time limit [34,35]. The resident population and floating populations are commonly found in the Chinese census, and the definitions are clear [36]. The floating population is a unique population group in China, which is closely related to the household registration system [37]. However, the definition of population type is disunity in terms of the big data. Wang put forward the definition of actual population, which was defined as the population that has stayed in the urban local space on average every day, and divided it into the static population and floating population [38]. Han and Shi further proposed time thresholds for defining population types [39,40]. Considering Han and Shi's classification of population types, we define the resident population as appearing in the same position at 21:00–07:00 for over 10 days in November. The floating population is defined as the users staying in the same city for more than 3 h in a day, but the nonresident population of the city. The total population is the sum of the resident population and the floating population.
- Population distribution characteristics. We divide the population into five levels, from high to low in 5,4,3,2,1. As Figure 3 shows, the resident population is mainly distributed within the sixth ring road and adjacent streets. The floating population is mainly distributed outside the fourth ring road, especially in Tongzhou, Daxing, and Changping, where there is a large floating population number. The distribution characteristics of the total population are roughly similar to the resident population, but there is also a significant increase in the population of some streets.

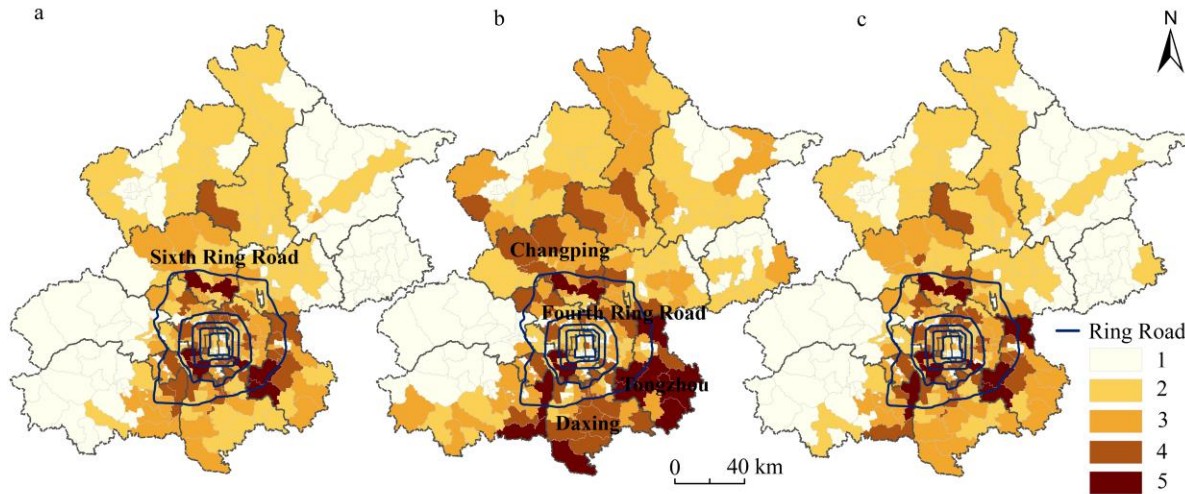

**Figure 3.** Distribution map of the population: (**a**) resident population; (**b**) floating population; (**c**) total population.

### 2.4. Methods

2.4.1. Kernel Density Estimation

Kernel density estimation is one of the most commonly effective methods in point pattern analysis. The essence of this method is a nonparametric analysis method [41].

The calculated results of this method show that the closer to the core, the greater the radiation value of the center, which conforms to the diffusion characteristics of the influence of the urban facility service on the surrounding location. Kernel density estimation has two advantages over other methods (such as the sample method). On the one hand, it can obtain continuous spatial characteristics. On the other hand, it considers the distance attenuation effect of the facility to its surrounding location services [42,43]. The method was used in our study to understand the agglomerate characteristics of hierarchical healthcare facilities. The calculation formula is:

$$f(x) = \sum_{i=1}^{1} \frac{1}{nh} k\left(\frac{d_{is}}{h}\right) \tag{1}$$

where $f(x)$ is the kernel density function; $h$ is the distance attenuation threshold; $n$ is the number of point elements within the search distance; $k$ is the spatial weight function; and $d_{is}$ is the distance from the center point $i$ to the point $s$ ($d_{is} < h$). The key to the kernel density method is the choice of bandwidth, which has a greater impact on the results. According to the Urban Public Service Facilities Planning Standards, the service radius of hospitals is 2000 m; the radius of primary healthcare institutions is 1000 m; and the radius of designated retail pharmacies is 1000 m.

### 2.4.2. Bivariate Spatial Autocorrelation Model

Bivariate spatial autocorrelation is an extension based on Moran's I index. Compared with other spatial analysis methods, the bivariate spatial autocorrelation model can accurately reflect the aggregation and differentiation characteristics of multiple spatial elements [44,45]. Assuming that there are two variables, A and B, we calculated the bivariate autocorrelation of A and B in a unified range of regions, and we can obtain four regional aggregation modes. The mean values of variable A and its adjacent block variable B are high, or the values of variable A and its adjacent block variable B are low. Moreover, the value of variable A is high and the mean value of variable B is low or the value of variable A is low and the mean value of variable B is high. The relationship between B and A is the same [46]. We used this model to find the spatial match of hierarchical healthcare facilities and residential areas in terms of quantity. The calculation formula is:

$$I_{ar}^i = Z_a^i \sum_{j=1}^{n} w_{ij} z_r^j \tag{2}$$

$$z_a^i = \frac{X_a^i - X_a}{\sigma_a} \tag{3}$$

$$z_r^i = \frac{X_r^j - X_r}{\sigma_r} \tag{4}$$

where $I_{ar}^i$ represents the bivariate local Moran spatial autocorrelation index; $w_{ij}$ is the weight of the spatial connection matrix between spatial units $i$ and $j$; $X_a^i$ is the kernel density of healthcare facilities in spatial unit $i$, $X_r^j$ is the kernel density of residential areas in spatial unit $j$; $z_a^i$ is the observation value standardized by the standard deviation of $X_a^i$; $z_r^j$ is the observation value standardized by the standard deviation of $X_r^j$; $X_a$ and $X_r$ are the average values of the kernel density of healthcare facilities and residential areas, respectively; $\sigma_a$ and $\sigma_r$ are the standard deviations of the kernel density of healthcare facilities and residential areas. $I_{ar}^i > 0$ indicates that the spatial positive correlation between the value of one observation variable in the region and the mean value of another variable in its neighborhood is high-high or low-low. $I_{ar}^i < 0$ indicates that the two variables have a low-high or high-low spatial negative correlation.

### 2.4.3. Geographical Concentration and Coupling Index

To further explore the spatial correlation between hierarchical healthcare facilities and the residential areas in terms of quantity, we adopted the geographical concentration and coupling index. Geographical concentration was first mentioned by Ellision, who explored the nature of agglomerative forces in describing patterns of concentration [47]. Now, it has been widely used to study the agglomeration degree of an economy and population in the space. Geographical concentration is an effective indicator to measure the spatial characteristics by comprehensively considering factors such as population, healthcare facilities, and area [48]. The calculation formula is:

$$Ri = \left(\frac{Ai}{\sum\limits_{i=1}^{n} Ai}\right) / \left(\frac{Ti}{\sum\limits_{i=1}^{n} Ti}\right) \tag{5}$$

where $R_i$ is the geographic concentration index of healthcare facilities or a population in street $i$; $A_i$ is the capacity of healthcare facilities or population in street $i$; and $T_i$ is the area of street $i$. The spatial match relationship is measured by the coupling index, which has outstanding performance in measuring the consistency of two variables [49,50]. The calculation formula is:

$$I = R_{healthcare\ facilities}/R_{population} \tag{6}$$

where $I$ is the coupling index; $R_{healthcare\ facilities}$ indicates the geographic concentration index of healthcare facilities in a street; $R_{population}$ represents the geographic concentration index of the population in a street. $I < 1$ means the lagging agglomeration of healthcare facilities; $I = 1$ indicates coordinated development of healthcare facilities agglomeration and population agglomeration; $I > 1$ means advanced agglomeration of healthcare facilities. Based on the division results in previous research [49,50], the coupling index $I$ is divided into four types: the lagged type of healthcare facilities ($0 < I < 0.5$), the developing type of healthcare facilities ($0.5 \leq I < 0.8$), the coordinated type of healthcare facilities ($0.8 \leq I < 1.5$), and the advanced type of healthcare facilities ($I \geq 1.5$). We used the total population as the capacity of the residential areas. Due to the differences in the service capacity of healthcare facilities at different levels, we took the annual average number of patients receiving treatment from different healthcare facilities multiplied by the number of healthcare facilities as the capacity. The annual average number of patients receiving treatment is shown in Table 2. Hospitals and primary healthcare institutions have official statistics on the annual average number of patients receiving treatment. However, designated retail pharmacies have no such data. On the one hand, it is difficult for designated retail pharmacies to obtain data on the behavior of personnel; on the other hand, it has no official classification, and the chance of being visited is equal for every pharmacy, so we used the same number of visits.

**Table 2.** The annual average number of patients receiving treatment.

| Healthcare Facility Level | Annual Average Number of Patients Receiving Treatment (Ten Thousand People) |
|---|---|
| Tertiary hospitals | 108.98 [1] |
| Secondary hospitals | 19.64 |
| Primary hospitals | 3.65 |
| Not evaluated | 1.66 |
| Community health service center(station) | 3.29 |
| Outpatient departments | 0.52 |
| Clinics | 0.18 |
| Health centers | 0.09 |
| Designated retail pharmacies | - |

[1] Data source: Statistics on Beijing's Health Work in 2019 (compendium).

## 3. Results

### 3.1. Analysis of Agglomeration Characteristics of Hierarchical Healthcare Facilities

3.1.1. Spatial Pattern Analysis of Hospitals

Figure 4 shows that the density of hospitals in the east is greater than in the west. High-density areas are mainly concentrated within the urban third ring road, and the agglomeration of hospitals in the eastern region is significantly higher than that in the western region. Hospitals are densely distributed along the line from Yonghe Palace Station to Tiantan East Gate Station of Subway Line 5 within the third ring road in the eastern region. The kernel density values of the adjacent Andingmen Street, Beixinqiao Street, Dongzhimen, Dongsi, Chaoyangmen, Jianguomen, and Chongmen Outside Street are at the highest level, between 1.38 and 2.66. However, there are only two high-value areas of kernel density in Xicheng within the third ring road in the western region, namely, Xinjiekou and Baizhifang. The areas outside the fourth ring road are mostly low-density areas, with only three high-density areas, namely Donghu, Tiantongyuan, and Chengbei. There are many blind areas with no hospitals outside the sixth ring road.

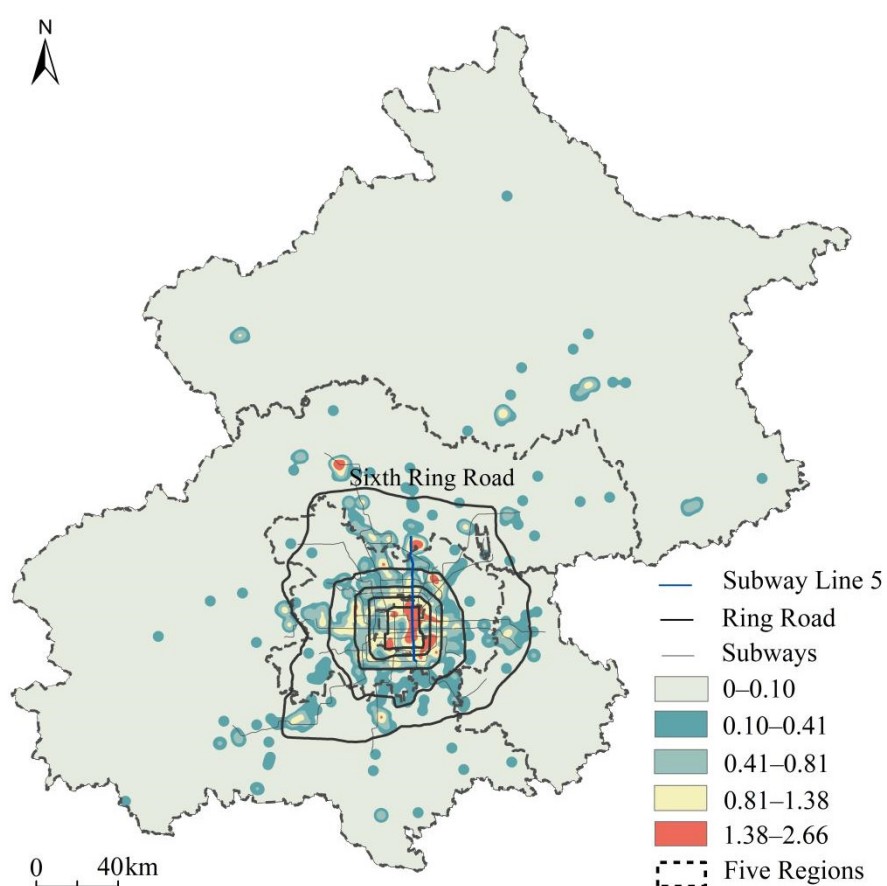

**Figure 4.** Distribution map of the kernel density of hospitals.

3.1.2. Spatial Pattern Analysis of Primary Healthcare Institutions

Figure 5 shows that primary healthcare institutions exhibit the spatial pattern of mass aggregation in the central city and multicentric dispersion in the urban peripheral. The high-density contiguous areas are mainly located in Dongzhimen, Chaoyangmen, Chaowai, Jianwai, and nearby. The phenomenon of inner block aggregation is closely related to the population density and the number of streets in the core urban area, which makes more community health service centers (stations) distribute in the core urban area with smaller areas. The outer multi-center discrete pattern is mainly affected by the number of health centers and clinics.

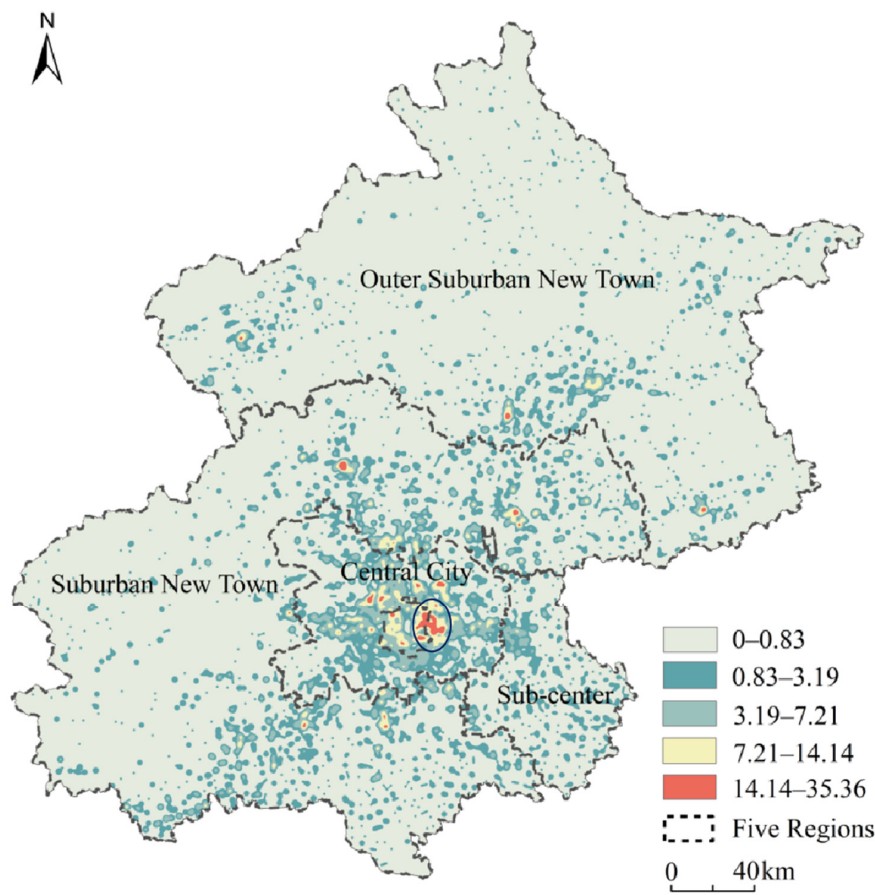

**Figure 5.** Distribution map of the kernel density of primary healthcare institutions.

### 3.1.3. Spatial Pattern Analysis of Designated Retail Pharmacies

Figure 6 shows that the designated retail pharmacies aggregate in the central urban area, and the cluster is small. The distribution of kernel density in Dongcheng, Xicheng, and Chaoyang is characterized by clusters. There are three high-value areas in Xicheng, two high-value areas in Dongcheng, and three high-value areas in Chaoyang. The aggregation effect of Haidian, Fengtai, and Shijingshan is low, with independent high-value areas. Although high-value areas appear in both the suburban and outer suburban new town, they are small in scale and do not form cluster aggregation patterns. The boundary of kernel density distribution is independent with an insufficient range of outward radiation. For example, in the Mentougou area, far away from the center of the area, there is almost no designated retail pharmacy coverage. Overall, designated retail pharmacies display the spatial characteristics of multi-core agglomeration, failing to form a system.

### 3.2. Analysis of Spatial Match from Quantity

Table 3 shows that all types of facilities have passed the significance test at the level of 1%. There is a high positive correlation between hierarchical healthcare facilities and residential areas, among which the hospitals agglomeration is largest, followed by primary healthcare institutions, and the designated retail pharmacies are smallest. Figure 7 shows the bivariate LISA diagram of three types of healthcare facilities and residential areas. High-high-agglomeration areas form one core multi-area spatial pattern. The one core of healthcare facilities at all levels is mostly distributed within the fourth ring line. The multi-area is mainly distributed in the central area outside the fourth ring line. Low-low-agglomeration areas are located in areas far away from the center of the sixth ring line, where the number of healthcare facilities is lower and the population density is low. Low-high-agglomeration areas are mainly distributed around high-high-concentration

areas, and designated retail pharmacies are the most distributed in the three types of facilities. High-low-agglomeration areas are rarely distributed. Similar to the results of kernel density aggregation, the high-high-agglomeration areas are located in core areas, and the low-low-agglomeration areas are located in the edge areas, fully showing the spatial heterogeneity of healthcare resources in quantitative terms.

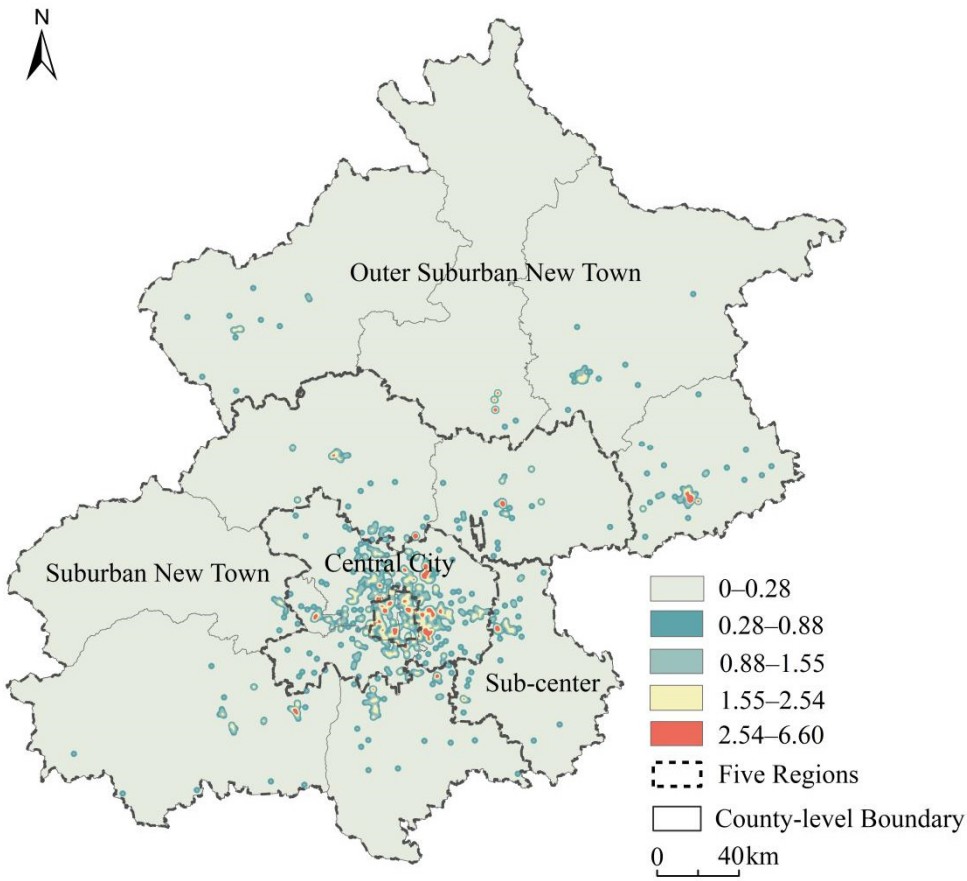

**Figure 6.** Distribution map of the kernel density of designated retail pharmacies.

**Table 3.** Results of bivariate spatial correlation analysis between healthcare facilities and residential areas.

| Category | Hospitals | Primary Healthcare Institutions | Designated Retail Pharmacies |
|---|---|---|---|
| Moran's I | 0.75 | 0.68 | 0.64 |
| *p*-value | 0.001 | 0.001 | 0.001 |
| Z-value | 425.76 | 396.40 | 388.80 |

*3.3. Analysis of Spatial Match from Capacity*

3.3.1. Spatial Match Analysis of Hospitals

From the coupling distribution map of the hospitals and the total population (Figure 8), it can be seen that the hospitals are highly intensive and unbalanced. The advanced type of hospitals is mainly located within the third ring road, which is consistent with the high-value area of the spatial distribution of hospital kernel density, mainly due to a large number of tertiary hospitals. From the proportion chart (Figure 8), the number of lagged types in hospitals account for 58.36%, and the overall coupling is inferior. The proportion of lagged types shows the characteristics of outer suburban new town > sub-center > suburban new town > central city > core area. Nine districts with the proportion of lagged type exceeding 50% are located in the sub-center, suburban, and outer suburban new town,

and the proportion of this type in Miyun is as high as 90%. The number of advanced and coordinated hospitals account for more than 50% of the four districts, Dongcheng, Xicheng, Haidian, and Shijingshan, which further show that there is a significant difference in urban areas. The layout of hospitals is not only regulated by government planning but also affected by historical development factors. The core area is still where the hospitals are concentrated, and the suburban and outer suburban new town generally lack hospitals.

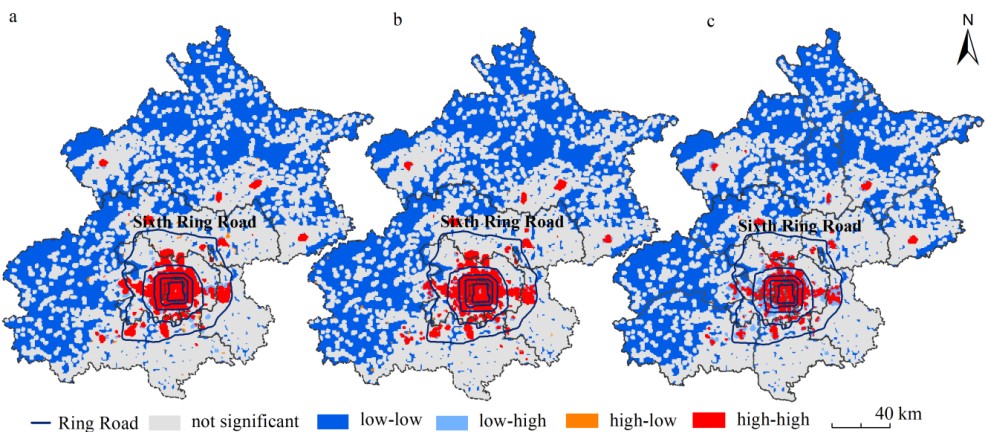

**Figure 7.** Bivariate LISA cluster map of healthcare facilities and residential areas: (**a**) hospitals; (**b**) primary healthcare institutions; (**c**) designated retail pharmacies.

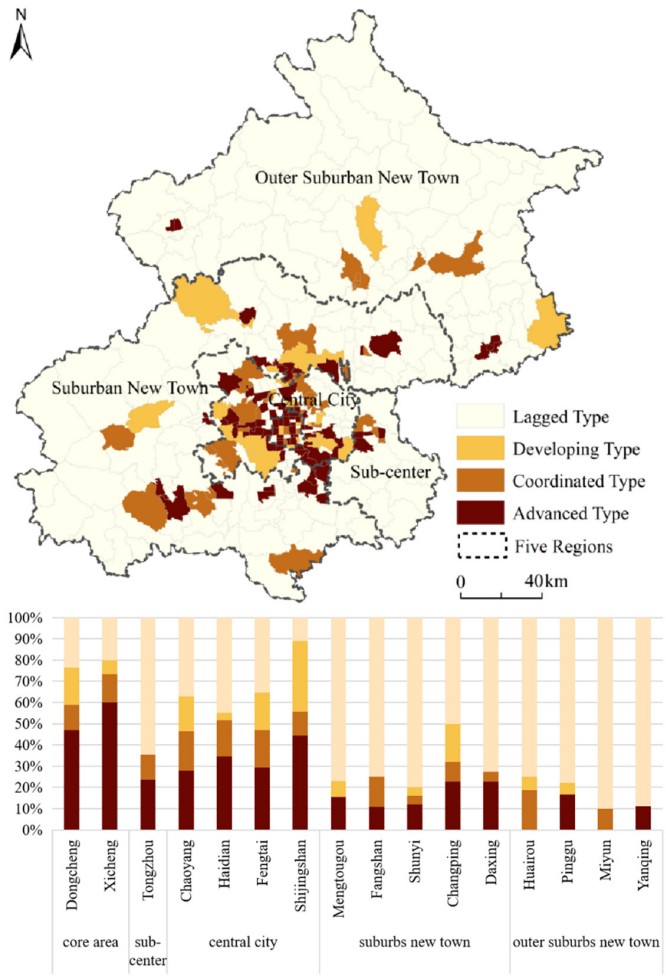

**Figure 8.** Matching types of hospitals.

### 3.3.2. Spatial Match Analysis of Primary Healthcare Institutions

The matching distribution map of primary healthcare institutions shows (Figure 9) that the matching of primary healthcare institutions and the total population is relatively superior. Compared with hospitals, advanced primary healthcare institutions mainly appear in Pinggu and Fangshan. The reasons are as follows. First, the population density of the outer suburban new town is relatively low. Second, the coverage of primary healthcare institutions is relatively wide. The outer suburban new town has achieved full coverage of health centers. From the diagram of the number of coupling types in each district (Figure 9), the primary healthcare institutions are mainly coordinated, accounting for 44.38%. The coordinated type shows the characteristics of central city > core area > suburban new town > outer suburban new town > sub-center. Among them, there are seven districts with the coordinated type accounting for more than 50%, covering all districts of the central city. The lagged types show the characteristics of sub-center > outer suburban new town > suburban new town > central city > core area. Except for the proportion of the sub-center being more than 40%, the remaining areas are less than 20%, and the proportion of the core area is 0. It is indicated that the level of spatial match in the sub-center is relatively insufficient, and the layout of primary healthcare institutions needs to be improved. Compared with the advanced types of hospitals, and the highly dense distribution patterns in core areas, the distribution of primary healthcare institutions is relatively homogeneous. For residents in the suburban and outer suburban new town, basic healthcare needs have been guaranteed.

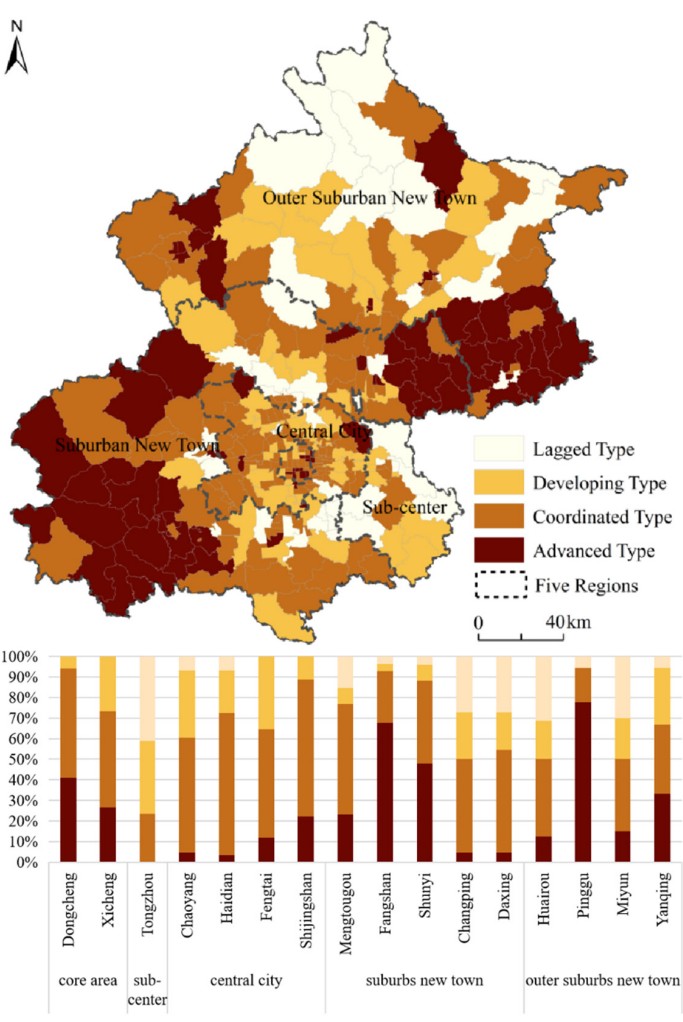

**Figure 9.** Matching types of primary healthcare institutions.

### 3.3.3. Spatial Match Analysis of Designated Retail Pharmacies

Designated retail pharmacies are the products of the healthcare insurance system, which mainly provide prescription dispensing and nonprescription drug retail services for the insured, among which the resident population is the main part. Therefore, in the coupling analysis, only the resident population is considered. According to Figure 10, the level of spatial match in Beijing is discordant. The lagged type of designated retail pharmacies includes most streets, accounting for 38.30% of the total. At the same time, there are great variants between regions. The advanced types and coordinated types of designated retail pharmacies are mainly distributed in core areas. The suburban and outer suburban new town are scattered, and some streets are even not covered by designated retail pharmacies. The main reasons are: first, under the background of the new healthcare reform, designated retail pharmacies are in the stage of gradual improvement and immature development; second, compared with ordinary retail pharmacies, designated retail pharmacies have a high threshold. Last, most of the designated retail pharmacies are group pharmacies and chain pharmacies. It is worth noting that, compared with other outer suburban new towns, Pinggu has a large number of advanced types, accounting for more than 60%, benefiting from the gathering of three chain pharmacies in Beijing Jiashitang Longxiang, Beijing Minbanghan, and Beijing Huayichun. In terms of the proportion of matching types in each district (Figure 10), the proportion of advanced types in Xicheng, Dongcheng, and Pinggu is more than 50%, while the proportion of lagged types in Mentougou and Huairou is also more than 50%. It shows that the designated retail pharmacies in the suburban and outer suburban new town have the worst matching with the resident population and cannot meet the needs of residents to purchase medicines nearby.

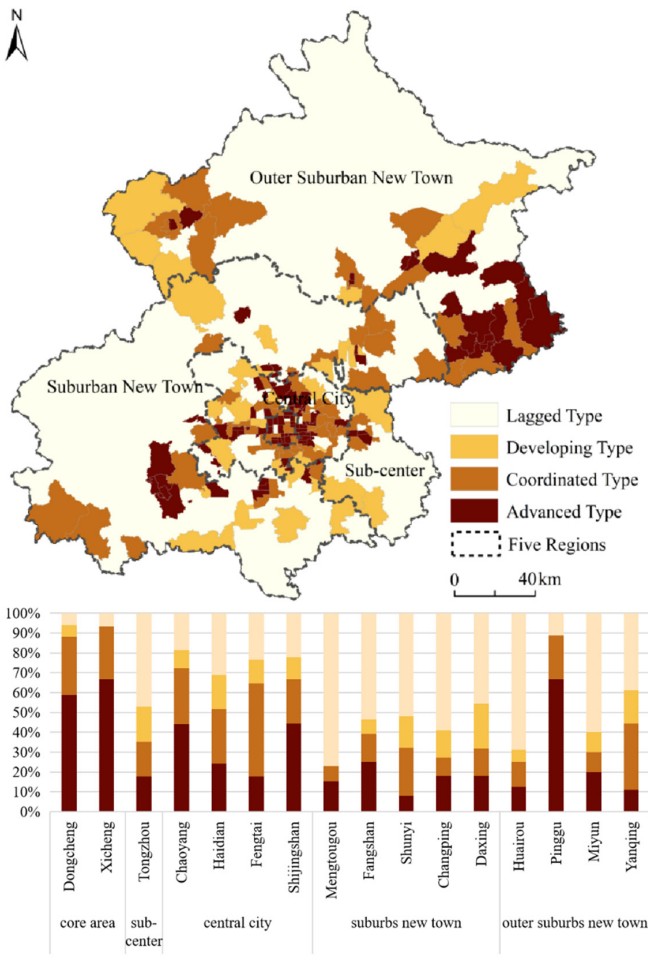

**Figure 10.** Matching types of designated retail pharmacies.

## 4. Discussion

### 4.1. Understanding Spatial Match from Quantity and Capacity

Early spatial pattern research by Zhao et al. [51] and Zhang et al. [28] has highlighted that the spatial patterns of healthcare resources in Beijing are extremely uneven. From the comparison of different urban areas, we find that hospitals at high aggregation are mainly concentrated in core areas, with a large number of high-level hospitals; by contrast, the number of hospitals in the peripheral urban area is small, and the problem of difficult access to medical services is relatively prominent. In addition, our results indicate that all types of healthcare facilities are positively correlated with residential areas. High-high-agglomeration areas are formed in core areas, and low-low-agglomeration areas are formed in marginal areas. This finding can further verify the characteristics of spatial patterns, also proved by previous studies [8]. Apart from the above two aspects, evidence is provided to highlight the designated retail pharmacies' spatial patterns and spatial match, seldom discussed in previous studies. Our results support the postulation that a well-organized hierarchical healthcare system should be designed from the perspective of different healthcare facilities [52].

Another important finding in this research is that a mismatched distribution is detected between healthcare facilities and the population at the street block level. We mainly describe two extremely matching types: population distribution intensity relative to the neighborhood without sufficient capacity of healthcare facilities; or conversely, the capacity of healthcare facilities in the area is much more intensive than that in surrounding areas whilst population distribution lags. The first type reflects the shortage of healthcare facilities' supply for large volumes of healthcare activities at the local scale. The second type suggests the surplus of healthcare facilities' supply. The first type occurs mostly in peripheral areas, while the second type occurs in core areas. Consequently, it is hard for residents in many peripheral areas to meet the need for nearby healthcare treatment. Moreover, growing healthcare activities are densely distributed in urban core areas, causing problems such as intensive healthcare and insufficient healthcare capacity. The gap between core areas and peripheral areas has been frequently reported [53,54]. This gap may be largely attributed to the policies of government management on the one hand, and the financial revenue on the other hand [41].

### 4.2. Policy Implications

The results from this study could inform healthcare layout and urban planning in other cities, as well as other public services facing equalization and demand for high-quality urban spaces. The revealed mismatch between population and healthcare facilities is important for solving the issue of inadequate healthcare services in Beijing. In recent years, Beijing has been implementing measures to relieve noncapital core functions. However, the progress of healthcare facilities is relatively slow. On the one hand, it has caused an excess and waste of healthcare resources in core areas, and on the other hand, it will also cause a shortage of healthcare resources in peripheric areas [54]. Due to the above situation, the regional distribution of healthcare resources should be adjusted to promote the equalization of high-quality resources. Combined with the ongoing noncapital function relief measures to maximize the existing healthcare resources in core areas, in terms of spatial distribution and facility service capabilities, the upward trend is inclined to promote the healthcare resources of peripheral areas. Besides, the government needs to strengthen the construction of designated retail pharmacies in the suburban and outer suburban new town through a variety of initiatives, focusing on the improvement of the overall level of healthcare services in the region. The spatial distribution and equalization of healthcare facilities are of great significance for eliminating spatial polarization and maintaining social equity. In consequence, the government should comprehensively consider regional balance from the aspects of spatial planning and policy formulation, coordinate various interests, take into account fairness and efficiency, and ensure rational and effective use of

medical resources, to weaken social contradictions, promote social harmony, and enhance the overall well-being of citizens.

### 4.3. Limitations and Prospects

The major limitation of this study is that the spatial match relationship is an abstract and complex concept concerning socio-economic attributes, personal preferences, government policies, and the reputation and quality of hospitals, which may be partly and incompletely measured by healthcare facilities and cellphone signaling data. Superior-quality hospitals will attract residents from wider regions, which may result in a lower actual matching of hospitals. Residents' healthcare conduct, which should also be an important aspect of spatial match, has not been included in this study, because of data unavailability. Thus, the results of the current study may be biased, with some findings attributed to data characteristics. Furthermore, the data of the annual average number of patients receiving treatment may not represent the real number of patients in each hospital. In future research, we plan to collect land use data and analyze the density of healthcare facilities in depth. Moreover, we will continue to explore how to use dynamic big data to integrate detailed information such as residents' behavior, and further deepen spatial match studies between healthcare facilities and residential areas. In addition, the number of patients receiving treatment data with detailed healthcare names and times might be an alternative for finding the patterns of patients receiving healthcare services and indicating spatial match. The proposed analytical framework in the study can be applied to other cities to examine spatial match patterns of public services.

### 5. Conclusions

The key findings are as follows. First, designated retail pharmacies display the characteristics of multi-core agglomeration in the central urban areas but do not form a large scale in the city. Second, in terms of quantity, hierarchical healthcare facilities are significantly positively correlated with the residential area. However, the quantitative matching patterns show that the spatial match of healthcare resources is unbalanced. High-high-concentration areas are all in core areas, and low-low-concentration areas are located in the urban periphery. Last, in terms of capacity, there are many diversities among hierarchical healthcare facilities. Hospitals are mainly lagged type, accounting for 58.36%. Hospitals within the third ring road are mainly advanced types, and the suburban and outer suburban new town are mainly lagged types. The primary healthcare institutions are mainly coordinated types, accounting for 44.38%, and the overall type is relatively homogeneous. The designated retail pharmacies are mainly lagged types, accounting for 38.30% of the total. Most of the suburban and outer suburban new town areas cannot meet the residents' demand for healthcare insurance to purchase medicines nearby. These results will further advance the system of hierarchical healthcare and fill the gap where traditional spatial matches ignore the designated retail pharmacies and the floating population. Practical insights from this study can assist in government policymaking and urban planning practices.

**Author Contributions:** The research is mainly conceived and designed by X.C. and H.W.; X.C. and Q.D. performed the experiments; X.C. wrote the manuscript; H.W., X.N. and P.J. reviewed the manuscript and provided comments. All authors have reviewed the manuscript. All authors have read and agreed to the published version of the manuscript.

**Funding:** This research is funded by the Fundamental Scientific Research Funds for Central Public Welfare Research Institutes (AR2117) and the Natural Resources Planning and Management Project (A2113).

**Institutional Review Board Statement:** Not applicable.

**Informed Consent Statement:** Not applicable.

**Data Availability Statement:** Not applicable.

**Conflicts of Interest:** The authors declare no conflict of interest.

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
