# Peer review of "Multidimensional Spatial Match of Hierarchical Healthcare Facilities Considering Floating Population: A Case of Beijing, China"

_sustainability, doi:10.3390/su14031092_

Round 1

Reviewer 1 Report

This research explores the spatial distributions of three categories of healthcare facilities (i.e. hospital, primary health care institution, and designated retail pharmacies) with the KDE and evaluates their spatial matches with the residential areas and ambient population. Overall, this paper is well-written, with the motivation and the method clearly introduced. The calculation of the coupling index and the reveal of two extreme matching types in Beijing are two major contributions of this paper. There are only a few minor issues that I would request further clarifications.

  1. In the abstract, the “high-high matching type” is not intuitively clear. Please rephrase it with a more understandable statement.

  1. It seems that the word “lag behind” is not appropriate and not easy to be understood.

  1. Section 2.2.2, line 131, remove the redundant sentence “. For subsequent analysis,”.

  1. Please explain more on how to calculate the capacity of healthcare facilities for each street. Has the service area of the healthcare facilities been considered?

  1. Section 3.1.3: “Small Overall Scale in Urban Areas”? I cannot understand it.

  1. Please describe the process of selecting the spatial unit of analysis?

  1. Please add a description of the spatial distribution of resident, floating, and total population

  1. Please give more introduction on the process of getting the residential areas.

Author Response

Dear reviewer,

Thank you very much for your kind comments on our manuscript. There is no doubt that these comments are valuable and very helpful for revising and improving our manuscript.Please see the attachment.

Reviewer 2 Report

  1. The abstract will need to be revised to convey the clear meaning and contribution of the study. Presently it does not clearly explain the spatial association pattern between healthcare facility and population, for example, ‘high-high or low-low matching types’ is not understandable. Likewise, ‘primary healthcare institution and population are mostly coordinated’ does not convey the meaning of a corresponding spatial association. 
  2. Font size of the variables in lines 174 to 181 must be fixed.
  3. The cellphone signaling data was collected for 10 days in November. Was it in 2021?
  4. More description of the mobile phone data is needed. For instance, how to clean the data. Descriptive statistics of the raw and cleaned cell phone signaling data will also be needed, possibly in Section 2.2.3.
  5. Section 2.2.3, the way the floating population is defined and extracted from the cell phone data should be reviewed and properly referenced.
  6. Floating population was defined in this study for those who stay for more than 3 hours a day. Proper reference to this definition is needed. Can it be any consecutive 3 hours in a day? Do they need to stay every day during the data collection period? A brief explanation of the algorithm will be useful.
  7. The classification of healthcare facilities is not clear. Each needs a proper definition. For instance, how significant is the designated retail pharmacy in the public health system? Table 1 may need to be improved. It is not clear which category name belongs to which data category, e.g., nursing homes are under hospitals or primary healthcare institutions? How much do we need for each kind of healthcare facility? Or can they be substitutional? For example, if the government provides more hospitals, we may not need many pharmacies. In addition, for hospitals, are they all the general hospitals? In some cases, there might be a specialized hospital. In that case will they be included or excluded from the data and analysis?
  8. To make the paper comprehensive, the literature review on the method will need to be enriched. Sections 2.3.1, 2.3.2, and 2.3.3 will need reference on fundamental and other existing studies or available methods. For example, density indicators rather than kernel density function; basic background on univariate Moran’s I prior to bivariate one; the other geographical concentration indices, etc.
  9. Figures 2 to 6 are small. Section 3.1.1, 3.1.2, and 3.1.3 mention various jurisdictions, streets, ring roads as well as subway line 5, but they could not be identified in the figures. These become less informative. The district names are in a very local context, so not conveying a meaningful message to the general reader. Suggest to extract the finding of these density analysis on some perspectives such as urban structure, urban agglomeration, land use or activity patterns, etc.
  10. Section 3.1.1, 3.1.2, and 3.1.3 titles should be concise. The phrases after colon must be a summary at the beginning or at the end of each subsection mentioning what has been found or concluded from the kernel density analysis. Again, mentioning the third ring road, inner block, outer multi-center, multi-core clusters will be more visualized by a map.
  11. Section 3.1.1 discusses the result at street/block level. However, this was not described in the methodology where the road network might need to be described.  
  12. Section 3.2 discusses many things that are supposed to be clear in Figure 3, e.g., the ring roads, agglomeration patterns, etc.  Again enlarging the figures with more information will help. More explanation for each type of the spatial association will be needed, e.g., what are the high (healthcare) - high (population) patterns judged, similarly will the low-how be problematic for all three types of the healthcare facilities?
  13. Relating to Figures 4 to 6, although the 16 areas that are categorized into core area, sub-center, central city in Section 2.1,  they will be more understandable to show in color, probably in Figure 1 too. Moreover, definition of each type of area will be useful, e.g., how the sub-center is defined, etc.
  14. It was not clear how the capacity of the designated retail pharmacies were defined and analyzed, i.e., Table 2 and Section 3.3.3. 

Author Response

(The authors gave the same response as above.)

Round 2

Reviewer 2 Report

  1. Adding the study flowchart in Figure 1 is helpful. As the result maps are presented in the subsequent sections, these are not needed here, apparently the text in the maps are too small. In addition, literally, the term quantitative analysis itself covers capacity analysis. Suggest to use a more appropriate term to reflect the analysis of the healthcare and population distribution.
  2. Although the references  were added in Section 2.4.1, 2.4.2, and 2.4.3 for the corresponding analysis methods, a bit further discussion on their findings, advantages or disadvantages, or suitability to the context being analyzed will make the paper more complete and meaningful.
  3. Population type identification can still be improved, lines 210 to 227. Adding more relevant references than [34,35] will be useful. 
  4. Figure 3, texts in white color are not very clear. 
  5. Recheck the meaning of the sentence in Line 266: we use this study to find…
